# Effects of blood urea nitrogen independent of the estimated glomerular filtration rate on the development of anemia in non-dialysis chronic kidney disease: The results of the KNOW-CKD study

Hyo Jin Kim[1,2], Tae Eun Kim[3], Miyeun Han[4], Yongin Yi[5], Jong Cheol Jeong[3,6], Ho Jun Chin[ORCID][3,6], Sang Heon Song[1,2], Joongyub Lee[7], Kyu-Beck Lee[8], Suah Sung[9], Seung Hyeok Han[10], Eun Young Seong[1,2], Curie Ahn[11], Kook-Hwan Oh[6,12], Dong-Wan Chae[ORCID][3,6]*

1 Department of Internal Medicine, Pusan National University Hospital, Pusan National University College of Medicine, Busan, Korea, 2 Biomedical Research Institute, Pusan National University Hospital, Busan, Korea, 3 Department of Internal Medicine, Seoul National University Bundang Hospital, Seongnamsi, Gyeonggi-do, Korea, 4 Department of Internal Medicine, Hallym University Hangang Sacred Heart Hospital, Seoul, Korea, 5 Department of Internal Medicine, Dankook University Hospital, Cheonan-si, Chungcheongnam-do, Korea, 6 Department of Internal Medicine, Seoul National University College of Medicine, Seoul, Korea, 7 Department of Preventive Medicine, Seoul National University College of Medicine, Seoul, Korea, 8 Department of Internal Medicine, Kangbuk Samsung Hospital, Sungkyunkwan University School of Medicine, Seoul, Korea, 9 Department of Internal Medicine, Eulji Medical Center, Eulji University, Seoul, Korea, 10 Department of Internal Medicine, Yonsei University College of Medicine, Seoul, Korea, 11 Department of Internal Medicine, National Medical Center, Seoul, Korea, 12 Department of Internal Medicine, Seoul National University Hospital, Seoul, Korea

* cdw1302@snubh.org

## Abstract

### Background

Anemia is a common complication of chronic kidney disease (CKD). Blood urea nitrogen (BUN) in CKD represents nitrogenous uremic toxin accumulation which could be involved in anemia of CKD. We investigated the effects of BUN independent of estimated glomerular filtration rate (eGFR) on anemia in non-dialysis CKD (NDCKD).

### Methods

This prospective study included 2,196 subjects enrolled in the KoreaN Cohort Study for Outcome in Patients With Chronic Kidney Disease (KNOW-CKD) cohort with BUN and hemoglobin level data. Initially, we investigated the association between BUN and hemoglobin level. To examine the impact of baseline BUN on the incident anemia, a longitudinal study was performed on 1,169 patients without anemia at study enrollment. BUN residuals were obtained from the fitted curve between BUN and eGFR. Anemia was defined as a hemoglobin level of <13.0 g/dL for men and <12.0 g/dL for women.

**Data Availability Statement:** All relevant data are within the paper and its Supporting information.

**Funding:** This study was supported by the Research Program funded by the Korea Center for Disease Control and Prevention (2011E3300300, 2012E3301100, 2013E3301600, 2013E3301601, 2013E3301602, 2016E3300200, 2016E3300201, 2016E3300202, 2019E320100, and 2019E320101). In addition, the authors thank the KNOW-CKD investigators, clinical research nurses, and the patients included in the study.

**Competing interests:** The authors have declared that no competing interests exist.

## Results

BUN residuals were not related to eGFR but to daily protein intake (DPI), while BUN was related to both eGFR and DPI. BUN was inversely associated with hemoglobin level (β -0.03; 95% confidence interval [CI] -0.04, -0.03; P <0.001) in the multivariable linear regression analysis adjusted for multiple confounders including eGFR, and BUN residual used instead of BUN was also inversely associated with hemoglobin level (β -0.03; 95% CI -0.04, -0.02; P <0.001). Among the 1,169 subjects without anemia at baseline, 414 (35.4%) subjects newly developed anemia during the follow-up period of 37.5 ± 22.1 months. In the multivariable Cox regression analysis with adjustment, both high BUN level (Hazard ratio [HR] 1.02; 95% CI 1.01, 1.04; P = 0.002) and BUN residual used instead of BUN (HR 1.02; 95% CI 1.00, 1.04; P = 0.031) increased the risk of anemia development. Moreover, BUN, rather than eGFR, increased the risk of anemia development in patients with CKD stage 3 in the multivariable Cox regression.

## Conclusion

Higher BUN levels derived from inappropriately high protein intake relative to renal function were associated with low hemoglobin levels and the increased risk of anemia independent of eGFR in NDCKD patients.

## Introduction

Anemia is a common complication of chronic kidney disease (CKD), the prevalence of which increases progressively as renal function declines [1]. Anemia leads to decreased quality of life, left ventricular hypertrophy, and mortality in CKD [2].

Although relative erythropoietin (EPO) deficiency and disordered iron metabolism are the major mechanisms of anemia of CKD, uremia-induced circulating inhibitors of erythropoiesis have also been reported to play some role in this disorder [3–6]. In a previous experimental study, primary bone marrow cells have shown dose-dependent growth inhibition when treated with the serum of uremic patients [7]. In another experimental study, acrolein (a uremic toxin) stimulated suicidal erythrocyte death [8]. Removal of furancarboxylic acid, an inhibitor of erythropoietin, by albumin-leaking hemodialysis (HD) significantly increased hematocrit in HD patients [9]. In this regard, it appears uremic toxins might influence anemia development in CKD patients although no specific uremic toxin(s) responsible for anemia of CKD has been identified yet.

Urea is a small water-soluble molecule that is freely filtered by the glomeruli and absorbed by the proximal and distal tubules of the kidney. Several urea transporters are involved in urea handling along the nephron [10]. Urea has quantitively the highest serum concentration among the different organic solutes retained in patients with CKD [11].

Blood urea nitrogen (BUN) is traditionally one of the indicators for evaluating kidney function, and BUN levels are inversely correlated with kidney function [12]. Besides glomerular filtration, BUN levels are also influenced by tubular resorption and production of urea. In acute illnesses, such as acute heart failure [13], acute pancreatitis [14], and extensive diuretic use [15], increased tubular reabsorption of urea consequent to neurohumoral activation originating from the depletion of effective circulating volume is a common cause of higher elevation of

BUN compared to glomerular filtration rate (GFR) leading to higher BUN/creatinine ratio in these patients.

Increased production of urea secondary to increased protein intake, is the most important extrarenal cause of elevation of BUN in stable CKD patients [16]. Different BUN levels were frequently observed in CKD patients having the same GFR [17] and the effect of protein intake on BUN became larger as estimated GFR (eGFR) declined in CKD [18]. Hence BUN in CKD is not only an indicator of kidney function but is also a marker of retention of nitrogenous uremic solutes mainly originating from protein intake. A very low-protein diet supplemented with essential amino acids reduced the EPO requirement to maintain the hemoglobin level in non-dialysis CKD (NDCKD) [19]. Increased urea reduction ratio increased hematocrit [20] or increased Kt/V urea reduced EPO requirement [21] in HD patients.

Therefore, we hypothesized that BUN would have independent effects of eGFR on the development of anemia in NDCKD by representing the accumulation of uremic toxins which could inhibit effective erythropoiesis in NDCKD. Because both BUN and hemoglobin level in CKD are strongly influenced by renal function represented by eGFR, a parameter which excludes the effect of eGFR on BUN is needed to evaluate the effect of BUN independent of eGFR on anemia in CKD. Residual was proposed as one of the methods modeling the association of a specific constituent and outcome [22] similar to our study in which eGFR, volume status, protein intake are important constituents of BUN. BUN residual was actually used in a study to assess the independent association of eGFR between BUN and the development of diabetes mellitus [23]. Considering that the distribution of BUN is widely spread at the same eGFR [17], the authors calculate BUN residual to minimize the impact of kidney function on BUN levels. BUN residuals were obtained from the fitted curve between BUN and eGFR to minimize the confounding effects from the strong collinearities between BUN, eGFR, and anemia [22, 23]. BUN residual may be an indicator that reflects factors affecting the BUN level except for the effects of eGFR, and in stable CKD patients, the count of protein intake will be the main factor in determining BUN residual. We also performed subgroup analyses in which the discordant effects of BUN and eGFR on the development of anemia could be found.

## Methods

### Study design and population

The KoreaN Cohort Study for Outcome in Patients With Chronic Kidney Disease (KNOW-CKD) is an ongoing Korean multicenter prospective cohort study conducted in patients with CKD stage 1–5. The detailed methods and design of the KNOW-CKD study have been previously described (NCT01630486 at http://www.clinicaltrials.gov) [24]. Briefly, patients aged between 20 and 75 years with various causes of CKD stages 1–5 were screened in each participating center from 2011 to 2016. In the initial study design of this prospective cohort, the authors had aimed to enroll a total of 2,450 patients. This number was calculated to include approximately 600 subjects in each subgroup (diabetic nephropathy, hypertensive nephropathy, glomerulonephritis, and polycystic kidney disease) with 80% power achieved to detect hazard ratios of approximately 2.00 and 1.60 for exposure with prevalence of 0.1 and 0.5, respectively, and a dropout rate of 10%. CKD was defined as eGFR <60 mL/min/1.73m$^2$ for ≥3 months by Chronic Kidney Disease Epidemiology Collaboration (CKD-EPI) creatinine equation [25] or presence of kidney damages, such as albuminuria, or pathologic or structural abnormalities detected by kidney biopsy or imaging study according to the Kidney Disease: Improving Global Outcomes (KDIGO) guidelines [26]. Of the 2,238 patients enrolled in the study from 2011 to 2016, we included 2,196 patients with BUN and hemoglobin data measured at enrollment (Fig 1). The protocol of the KNOW-CKD study was approved in 2011 by the

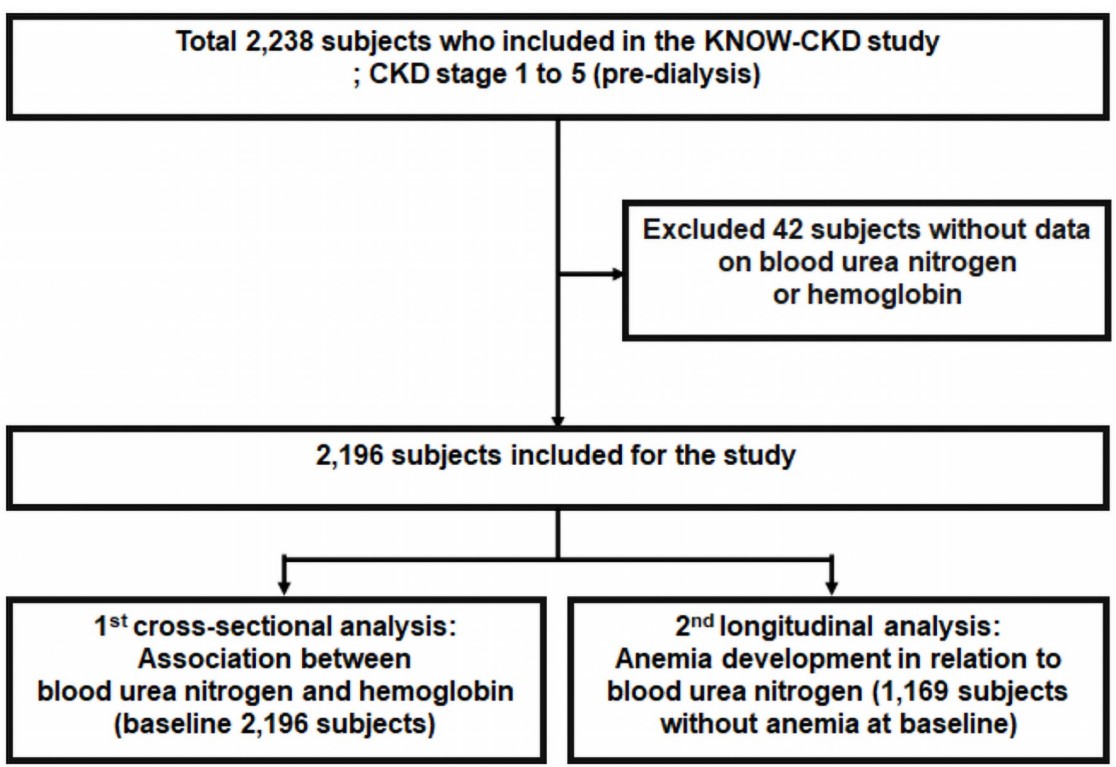

**Fig 1. Flow chart of subject enrollment and analyses.**

ethical committees of all participating clinical centers, that is, the Institutional Review Boards of Seoul National University Hospital (1104-089-359), Seoul National University Bundang Hospital (B-1106/129-008), Yonsei University Severance Hospital (4-2011-0163), Seoul St. Mary's Hospital (KC11OIMI0441), Kangbuk Samsung Medical Center (2011-01-076), Eulji General Hospital (201105–01), Gil Hospital (GIRBA2553), Chonnam National University Hospital (CNUH-2011-092), and Pusan Paik Hospital (11–091). This study was performed in accordance with the principles of the Declaration of Helsinki and all study subjects provided written informed consent.

## Clinical and laboratory measurements

Clinical characteristics, such as demographic factors, laboratory values, and medications, were extracted from an electronic data management system (http://www.phactax.org). Hypertension (HTN) was defined as a systolic blood pressure (SBP) of $\geq$140 mmHg or diastolic blood pressure of $\geq$90 mmHg, a history of HTN, or taking antihypertensive drugs. Diabetes mellitus (DM) was defined as a fasting serum glucose of $\geq$126 mg/dL, a history of DM, or taking insulin or oral antidiabetic drugs. The laboratory variables were measured using $\geq$8-hour fasting blood samples. The laboratory tests including complete blood count and blood chemistry were performed at baseline, at 6 and 12 months after study commencement, and annually thereafter. The serum creatinine level was measured using an isotope dilution mass spectrometry (IDMS)-traceable method [27] at a central laboratory (Lab Genomics, Seoul, Korea). eGFR was calculated using the CKD-EPI creatinine equation [25]. Stage 1–5 CKD was defined according to the KDIGO guidelines [26]. Anemia was defined as a hemoglobin level of <13.0 g/dL for men and <12.0 g/dL for women using the World Health Organization criteria [28].

Transferrin saturation (TSAT) was calculated by dividing the serum iron level by the total iron-binding capacity. Iron deficiency was defined as a ferritin level of ≤100 ng/mL and a TSAT of ≤20% [29]. Intravenous or oral iron supplementation was prescribed under the discretion of physicians in participating hospitals if physicians judged it necessary to raise the hemoglobin of his patient or to reduce the dose of ESA in accordance with the KDIGO guidelines [30]. Erythropoiesis-stimulating agent (ESA) was prescribed under the discretion of physicians in participating hospitals in patients with anemia of CKD with hemoglobin < 10 g/dL in accordance with the KDIGO guidelines [30], excluding other causes of anemia. In addition, ESA was started in patients with eGFR < 30 mL/min/1.73m$^2$ in accordance with the insurance coverage of Korea. During the use of ESA, physicians adjusted the dose of ESA to maintain hemoglobin at 10–11 g/dL (or hematocrit 30–33%) and stopped ESA if hemoglobin was > 11 g/dL (or hematocrit > 33%).

Second voided or random urine samples were immediately sent to a central laboratory to determine urine creatinine and protein levels. The urinary protein excretion was quantified using random urinary protein-to-creatinine ratio (UPCR, g/g). Twenty-four hours urine was also collected. The completeness of 24 hours urine collection was determined by a ratio of the measured 24 hours urine creatinine to the expected 24 hours urine creatinine, which was calculated using the Tanaka equation [31]. Estimated daily protein intake (DPI) was estimated using the Maroni-Mitch formula: 6.25 × [urine urea nitrogen (g/day) + 0.03 × ideal body weight (kg)] + proteinuria (g/day) [32], and DPI was calculated by dividing the estimated DPI by actual body weight (g/kg/day). The serum hepcidin level was measured at a central laboratory via cELISA using EIA5258 kits (DRG Diagnostics, Marburg, Germany), according to the manufacturer's instructions.

## Primary outcome

Initially, we investigated the association between BUN and hemoglobin levels in the 2,196 patients at baseline. To examine the impact of baseline BUN level on incident anemia, a longitudinal study was performed on 1,169 patients who did not have anemia at study enrollment. The primary outcome of the present study was the onset of *de novo* anemia development during follow-up.

## Statistical analyses

Continuous variables were presented as mean ± standard deviation (SD) or as median (interquartile range) and compared using one-way analysis of variance (ANOVA) or Kruskal-Wallis test in Table 1 and *t*-test or Mann-Whitney test in Table 3. Categorical variables were presented as the frequencies and percentages and analyzed using the Chi-square test or Fisher's exact test as appropriate. The *P*-value for the trend, which represents the dose-dependent effect of BUN on variables, was obtained using the ANOVA linear contrasts or Jonckheere-Terpstra test (continuous variables) or linear by linear association (categorical variables) in Table 1. Log transformation was used to normalize variables with a skewed distribution. To remove the effect of eGFR on BUN, a fitted curve between eGFR and BUN was obtained using the "splines" packages in R Statistics (version 4.04) which calculated the expected BUN value corresponding to eGFR using prediction function. The BUN residual was obtained by the vertical distance from each BUN value to the fitted curve between eGFR and BUN (Fig 2) [22, 23]. Multivariable linear regression analysis was performed to find the factors related to BUN and BUN residual. Multivariable linear regression analysis was also conducted after adjusting for the multiple confounders to investigate the association between BUN and serum hemoglobin levels; BUN residual was used instead of BUN in this analysis also. These models were

**Table 1. Clinical characteristics of the study subjects at enrollment stratified by BUN levels.**

| Characteristics | Total (N = 2,196) | Blood urea nitrogen | | | | P-value | [††]P for trend |
|---|---|---|---|---|---|---|---|
| | | 1st quartile (5.0–17.0 mg/dL) (n = 572) | 2nd quartile (17.1–23.9 mg/dL) (n = 520) | 3rd quartile (24.0–35.0 mg/dL) (n = 557) | 4th quartile (35.1–112.0 mg/dL) (n = 547) | | |
| Age (mean ± SD) | 53.7 ± 12.3 | 47.5 ± 12.0 | 54.1 ± 11.7 | 56.3 ± 11.8 | 57.0 ± 11.1 | <0.001 | <0.001 |
| Sex, male, n (%) | 1339 (61.0) | 326 (57.0) | 327 (62.9) | 347 (62.3) | 339 (62.0) | 0.155 | 0.111 |
| SBP (mmHg) | 127.8 ± 16.2 | 126.5 ± 14.6 | 127.1 ± 15.6 | 127.0 ± 15.9 | 130.8 ± 18.4 | <0.001 | <0.001 |
| BMI (kg/m$^2$) | 24.6 ± 3.4 | 24.5 ± 3.6 | 24.8 ± 3.4 | 24.5 ± 3.2 | 24.5 ± 3.3 | 0.484 | 0.675 |
| DM, n (%) | 743 (33.9) | 103 (18.1) | 140 (27.0) | 226 (40.6) | 274 (50.1) | <0.001 | <0.001 |
| HTN, n (%) | 2109 (96.1) | 519 (90.7) | .505 (97.1) | 544 (97.8) | 541 (98.9) | <0.001 | <0.001 |
| Age adjusted CCI | 3.4 ± 2.2 | 1.6 ± 1.8 | 3.3 ± 2.0 | 4.3 ± 1.8 | 4.6 ± 1.7 | <0.001 | <0.001 |
| Smoking status, n (%) | | | | | | 0.027 | 0.003 |
| Never | 1184 (53.9) | 334 (58.4) | 281 (54.1) | 299 (57.7) | 270 (49.4) | | |
| Current or former | 1011 (46.1) | 238 (41.6) | 238 (45.9) | 258 (46.3) | 277 (50.6) | | |
| CKD stage, n (%) | | | | | | <0.001 | <0.001 |
| Stage 1 | 354 (16.1) | 298 (52.1) | 48 (9.2) | 7 (1.3) | 1 (0.2) | | |
| Stage 2 | 414 (18.9) | 199 (34.8) | 163 (31.3) | 44 (7.9) | 8 (1.5) | | |
| Stage 3a | 360 (16.4) | 55 (9.6) | 189 (36.3) | 109 (30.3) | 7 (1.3) | | |
| Stage 3b | 458 (20.9) | 17 (3.0) | 104 (20.0) | 248 (44.5) | 89 (16.3) | | |
| Stage 4 | 473 (21.5) | 2 (0.3) | 16 (3.1) | 143 (25.7) | 312 (57.0) | | |
| Stage 5 | 137 (6.2) | 1 (0.7) | 0 (0.0) | 6 (1.1) | 130 (23.8) | | |
| Cause of CKD | | | | | | <0.001 | 0.012 |
| GN | 785 (35.7) | 250 (41.3) | 215 (41.3) | 176 (31.6) | 144 (26.3) | | |
| DN | 508 (23.1) | 45 (7.9) | 81 (15.6) | 161 (28.9) | 221 (40.4) | | |
| Hypertensive nephropathy | 405 (18.4) | 68 (11.9) | 96 (18.5) | 129 (23.2) | 112 (20.5) | | |
| ADPKD | 362 (16.5) | 173 (30.2) | 98 (18.8) | 53 (9.5) | 38 (6.9) | | |
| Others | 136 (6.2) | 36 (6.3) | 30 (5.8) | 38 (6.8) | 32 (5.9) | | |
| Creatinine (mg/dL) | 1.8 ± 1.2 | 0.9 ± 0.4 | 1.3 ± 0.4 | 1.9 ± 0.6 | 3.1 ± 1.4 | <0.001 | <0.001 |
| eGFR(mL/min/1.73m$^2$) | 53.0 ± 30.8 | 88.9 ± 23.4 | 59.5 ± 20.1 | 39.9 ± 15.4 | 22.7 ± 11.1 | <0.001 | <0.001 |
| BUN (mg/dL) | 28.3 ± 15.8 | 13.6 ± 2.6 | 20.4 ± 1.9 | 28.9 ± 3.3 | 50.6 ± 14.1 | <0.001 | <0.001 |
| BUN residual (mg/dL) | 0 ± 8.6 | -3.0 ± 4.1 | -1.8 ± 5.2 | -1.1 ± 7.1 | 5.9 ± 12.4 | <0.001 | <0.001 |
| [*]DPI, median, (Q1, Q3) (g/kg/day) | 0.99 (0.84, 1.18) | 0.99 (0.82, 1.16) | 1.00 (0.86, 1.22) | 0.99 (0.83, 1.18) | 0.98 (0.84, 1.16) | 0.222 | 0.930 |
| WBC (×10$^3$/mm$^3$) | 6611 ± 1916 | 6397 ± 1882 | 6495 ± 1827 | 6851 ± 2014 | 6701 ± 1905 | <0.001 | <0.001 |
| Platelet (×10$^3$/mm$^3$) | 231 ± 62 | 237 ± 59 | 229 ± 62 | 230 ± 61 | 227 ± 65 | 0.020 | 0.008 |
| Hemoglobin (g/dL) | 12.8 ± 2.0 | 14.0 ± 1.6 | 13.6 ± 1.8 | 12.5 ± 1.9 | 11.2 ± 1.5 | <0.001 | <0.001 |
| Iron (ug/dL) | 92.6 ± 35.1 | 102.1 ± 39.2 | 96.4 ± 34.4 | 88.7 ± 33.4 | 83.2 ± 29.9 | <0.001 | <0.001 |
| Ferritin (pmol/L) | 99.7 (53.7, 176.3) | 88.8 (42.9, 161.4) | 91.6 (53.3, 174.4) | 96.9 (58.9, 176.6) | 115.2 (63.7, 196.0) | <0.001 | <0.001 |
| TSAT (%) | 31.7 ± 12.1 | 32.9 ± 13.2 | 32.0 ± 11.7 | 31.2 ± 11.7 | 30.6 ± 11.4 | 0.011 | 0.001 |
| Anemia (%) | 970 (44.2) | 75 (13.1) | 137 (26.3) | 303 (54.4) | 455 (83.2) | <0.001 | <0.001 |
| Iron deficiency, n (%) | 1183 (54.9) | 316 (56.7) | 292 (27.4) | 308 (56.3) | 267 (49.4) | 0.030 | 0.017 |
| ESA use, n (%) | 167 (7.6) | 0 (0) | 4 (0.8) | 32 (5.8) | 131 (24.0) | <0.001 | <0.001 |
| Iron supplement, n (%) | 322 (14.7) | 20 (3.5) | 49 (9.5) | 85 (15.4) | 168 (30.8) | <0.001 | <0.001 |
| Albumin (g/dL) | 4.2 ± 0.4 | 4.3 ± 0.4 | 4.2 ± 0.4 | 4.1 ± 0.5 | 4.1 ± 0.4 | <0.001 | <0.001 |
| Phosphorus (mg/dL) | 3.7 ± 0.7 | 3.5 ± 0.5 | 3.5 ± 0.5 | 3.7 ± 0.6 | 4.2 ± 0.8 | <0.001 | <0.001 |
| Uric acid (mg/dL) | 7.0 ± 1.9 | 5.9 ± 1.6 | 6.8 ± 1.6 | 7.4 ± 1.8 | 8.1 ± 2.0 | <0.001 | <0.001 |
| Total CO$_2$ (mmol/L) | 25.7 ± 3.7 | 27.7 ± 3.0 | 26.9 ± 3.0 | 25.1 ± 3.2 | 23.0 ± 3.5 | <0.001 | <0.001 |
| Sodium (mmol/L) | 140.8 ± 2.5 | 140.7 ± 2.3 | 141.0 ± 2.2 | 141.0 ± 2.7 | 140.6 ± 2.7 | 0.016 | 0.680 |

*(Continued)*

**Table 1.** (Continued)

| Characteristics | Total (N = 2,196) | Blood urea nitrogen | | | | P-value | [††]P for trend |
|---|---|---|---|---|---|---|---|
| | | 1st quartile | 2nd quartile | 3rd quartile | 4th quartile | | |
| | | (5.0–17.0 mg/dL) | (17.1–23.9 mg/dL) | (24.0–35.0 mg/dL) | (35.1–112.0 mg/dL) | | |
| | | (n = 572) | (n = 520) | (n = 557) | (n = 547) | | |
| Potassium (mmol/L) | 4.6 ± 0.6 | 4.3 ± 0.4 | 4.4 ± 0.4 | 4.7 ± 0.6 | 5.0 ± 0.6 | <0.001 | <0.001 |
| Total cholesterol (mg/dL) | 174.1 ± 39.3 | 177.4 ± 35.1 | 177.7 ± 39.8 | 173.0 ± 40.6 | 168.4 ± 40.9 | <0.001 | <0.001 |
| LDL cholesterol (mg/dL) | 96.9 ± 31.9 | 101.0 ± 31.4 | 99.6 ± 31.2 | 95.3 ± 32.8 | 91.9 ± 31.3 | <0.001 | <0.001 |
| Triglyceride (mg/dL) | 157.1 ± 98.4 | 145.6 ± 82.8 | 156.8 ± 101.7 | 165.2 ± 104.7 | 161.2 ± 102.7 | 0.007 | 0.004 |
| HDL cholesterol (mg/dL) | 49.3 ± 15.4 | 53.3 ± 15.8 | 50.5 ± 15.4 | 47.8 ± 14.3 | 45.4 ± 14.9 | <0.001 | <0.001 |
| CRP, median, (Q1, Q3) (mg/L) | 0.6 (0.2, 1.7) | 0.5 (0.2, 1.5) | 0.6 (0.2, 1.4) | 0.7 (0.3, 1.9) | 0.7 (0.3, 2.0) | <0.001 | <0.001 |
| UPCR, median, (Q1, Q3) (g/g) | 0.5 (0.1, 1.5) | 0.2 (0.1, 0.7) | 0.3 (0.1, 1.0) | 0.6 (0.2, 2.0) | 1.1 (0.4, 2.6) | <0.001 | <0.001 |
| Hepcidin (ng/mL) | 12.9 (6.5, 23.8) | 8.9 (4.5, 16.7) | 11.0 (6.1, 19.1) | 14.2 (7.2, 24.9) | 21.1 (10.8, 37.1) | <0.001 | <0.001 |
| [**]24hr-urine volume, median, (Q1, Q3) (ml/day) | 1870 (1500, 2350) | 1800 (1400, 2206) | 1840 (1495, 2300) | 1900 (1500, 2325) | 1960 (1630, 2500) | <0.001 | <0.001 |
| ACEi or ARB, n (%) | 1874 (85.3) | 477 (83.4) | 454 (87.3) | 476 (85.5) | 467 (85.4) | 0.341 | 0.513 |
| [†]Diuretics, n (%) | 651 (30.3) | 70 (12.6) | 108 (21.4) | 198 (36.0) | 275 (50.9) | <0.001 | <0.001 |
| Statin, n (%) | 1133 (51.6) | 248 (21.9) | 272 (24.0) | 321 (28.3) | 292 (25.8) | <0.001 | <0.001 |

[††]P for the trend representing the dose-dependent effect of BUN on variables was obtained using the one-way analysis of variance linear contrasts or Jonckheere-Terpstra test (continuous variables) or linear by linear association (categorical variables).

SD, standard deviation; SBP, systolic blood pressure; BMI, body mass index; DM, diabetes mellitus; HTN, hypertension; CCI, Charlson comorbidity index; GN, glomerulonephritis; DN, diabetic nephropathy; ADPKD, autosomal dominant polycystic kidney disease; eGFR, estimated glomerular filtration rate as determined by the CKD-EPI creatinine equation; BUN, blood urea nitrogen; DPI, dietary protein intake; WBC, white blood cell; TSAT, transferrin saturation; ESA, erythropoiesis-stimulating agent; LDL, low-density lipoprotein; HDL, high-density lipoprotein; CRP, C-reactive protein; UPCR, urinary protein-to-creatinine ratio; ACEi, angiotensin converting enzyme inhibitor; ARB, angiotensin II receptor blocker

[*]Only 1522 and [**]1627 patients in whom 24hr-urine were properly collected and DPI could be calculated were included in the analysis.

[†]Loop or distal tubule diuretics

constructed after rigorous and stepwise adjustments for confounding factors. Model 1 was unadjusted model with no covariables. Model 2 was adjusted for demographic information and comorbidities data including age, sex, smoking, SBP, DM, HTN, and Charlson comorbidity index (CCI). Model 3 included laboratory parameters of eGFR, UPCR, white blood cell (WBC), platelet, C-reactive protein (CRP), albumin, total cholesterol, ferritin, hepcidin, iron, and TSAT in addition to the variables adjusted in model 2. Model 4 further included medications such as iron replacement therapy, ESA, angiotensin converting enzyme inhibitor (ACEi) or angiotensin II receptor blocker (ARB), statin, and diuretics. The data of the 1,169 (53.2%) patients without anemia at enrollment were subjected to further analysis to determine whether baseline BUN can predict the future development of anemia. Cox proportional hazard models adjusted for confounders were used to determine whether the continuous and categorical values of BUN were associated with anemia development. The results were presented as hazard ratios (HRs) and 95% confidence intervals (CIs). The same analysis was also performed using BUN residual instead of BUN to determine whether the continuous and categorical values of BUN residual were associated with anemia development. For each analysis, aforementioned four different sequential adjust models were used. A cubic spline analysis was used to evaluate any non-linear relationship between BUN or BUN residual levels as continuous variable and the risk of anemia development after adjusting for the confounders. Subgroup analyses were also performed to find the discordant effects of BUN and eGFR on the development of anemia

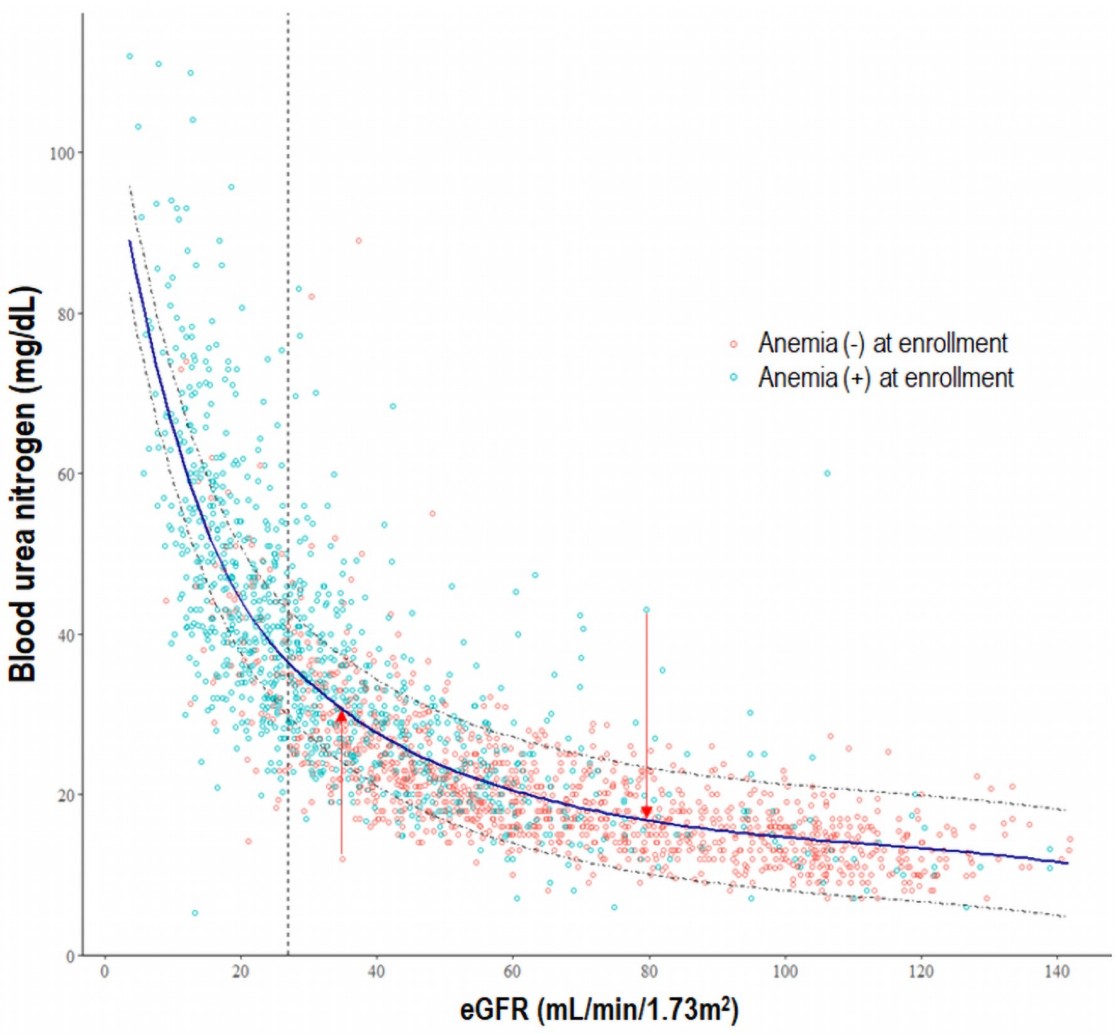

**Fig 2. Relationship between eGFR and BUN.** The BUN residuals were obtained from the regression curve between baseline BUN and eGFR levels to examine the effect of BUN on anemia development uncorrelated with eGFR. The solid line represents the fitted curve between BUN and eGFR. The BUN residuals were calculated as the vertical distance from each observation point to the fitted curve (arrows). Even at the same eGFR level, BUN concentration demonstrated a wide range (exemplified by a vertical dotted line). The dotted curve represents ± 0.5 × standard deviation from the fitted curve. The blue circles represent patients who had anemia at enrollment and the red circles denote patients who did not have anemia at enrollment. BUN, blood urea nitrogen; eGFR, estimated glomerular filtration rate as determined by the CKD-EPI creatinine equation.

according to the CKD stages. A *P*-value of <0.05 was considered statistically significant. The "survival and splines" R software package (version 4.04) was used to obtain BUN residual, and restricted cubic spline curves. SPSS statistical software (SPSS version 20.0, IBM Co., Armonk, NY, USA) was used for the other statistical analyses.

## Results

### Baseline clinical characteristics of the study subjects

The clinical characteristics of the 2,196 subjects by serum BUN quartile are presented in Table 1. The mean age was 53.7 ± 12.3 years, and 1,339 (61.0%) of the patients were men. The mean eGFR was 53.0 ± 30.8 mL/min/1.73m$^2$. The prevalence of DM (*P* <0.001) and HTN

(*P* <0.001) were higher in the high BUN quartiles. eGFR (*P* <0.001) was lower and the BUN residual (*P* <0.001) was higher in the high BUN quartiles. However, DPI was not significantly different according to the BUN quartiles (*P* = 0.222). The mean hemoglobin level was 12.8 ± 2.0 g/dL and was lower in the high BUN quartiles (*P* <0.001). In addition, serum iron (*P* <0.001) and TSAT (*P* = 0.011) levels were also lower in the high BUN quartiles. The serum ferritin was higher in the high BUN quartiles (*P* <0.001). As anemia was more prevalent in the high BUN quartiles, subjects in the high quartiles received more ESA and iron supplement therapy (*P* <0.001 for both). Subjects in the high BUN quartiles received more diuretics (*P* <0.001). Considering the types and dose of diuretics, the actual dose of the diuretics was not so high (S1 Table).

## Blood urea nitrogen residual

The scatter plot derived from baseline eGFR and BUN of the individual patients (Fig 2) demonstrated that BUN was inversely associated with eGFR, and BUN showed a range of different values at the same eGFR. The slope of the fitted curve between BUN and eGFR obtained from this scatter plot rapidly increased as eGFR decreased. The BUN residuals were calculated as the vertical distance from each observation point to the fitted curve. The BUN residual was higher in high BUN quartiles (Table 1; *P* <0.001). This scatter plot also demonstrated that the frequency of anemia increased as eGFR decreased. Interestingly, the frequency of anemia in patients with the same eGFR differed widely depending on their BUN values. The subjects with BUN value above the fitted curve (i.e., higher BUN residual) had anemia more frequently than those with BUN value below the curve (i.e., lower BUN residual) despite having the same eGFR value. As shown in S2 Table, BUN residual was not significantly related to eGFR or CKD stage, while BUN showed significant associations with eGFR (*P* <0.001) and CKD stage in the multivariable linear regression analysis. Interestingly, DPI was the main determinant of BUN and BUN residual in the same analysis. Sex, daily urine volume, statin usage, and diuretic usage were associated with both BUN and BUN residual. The constituents of dietary protein such as phosphate, uric acid, and total $CO_2$ content which is affected by acid content in protein were also related to both BUN and BUN residual. Importantly, there was no difference in the significant variables determining BUN and BUN residual except eGFR and CKD stage.

## Cross-sectional analysis of the association between BUN, BUN residual, and hemoglobin level

Table 2 summarizes the results of the linear regression analysis for the association between BUN, BUN residual, and hemoglobin level. The hemoglobin level was inversely associated with BUN in the unadjusted model (*β* -0.07; 95% CI -0.07, -0.06; *P* <0.001), and this association persisted after sequential adjustments for confounders (models 2–4). In the model fully adjusted for age, sex, smoking, SBP, DM, HTN, CCI, eGFR, log UPCR, WBC, platelet, platelet, log CRP, albumin, total cholesterol, log ferritin, log hepcidin, iron, TSAT, iron replacement therapy, ESA, ACEi or ARB, statin, and diuretics, the hemoglobin level was significantly associated with BUN (*β* -0.03; 95% CI -0.04, -0.03; *P* <0.001). The hemoglobin level was also inversely associated with BUN residual in the unadjusted model (*β* -0.02; 95% CI -0.03, -0.01; *P* < 0.001), and this association persisted after sequential adjustments for confounders (models 2–4). In the fully adjusted model (model 4), hemoglobin was significantly associated with the BUN residual (*β* -0.03; 95% CI -0.04, -0.02; *P* <0.001). When adjustment was done for the types and dose of diuretics and not just the use of diuretics (group 1: none; group 2: hydrochlorothiazide ≤ 12.5 mg, indapamide ≤ 1.5 mg, metolazone ≤ 2.5 mg; group 3: hydrochlorothiazide > 12.5 mg, indapamide > 1.5 mg, metolazone > 2.5 mg; group 4:

**Table 2. Multivariable linear regression analyses of the association between BUN, BUN residual, and hemoglobin level.**

|  | Model 1 | | Model 2 | | Model 3 | | Model 4 | |
|---|---|---|---|---|---|---|---|---|
|  | β (95% CI) | P value | β (95% CI) | P value | β (95% CI) | P value | β (95% CI) | P value |
| **BUN (per 1 mg/dL)** | -0.07 (-0.07, -0.06) | <0.001 | -0.06 (-0.06, -0.05) | <0.001 | -0.04 (-0.04, -0.03) | <0.001 | -0.03 (-0.04, -0.03) | <0.001 |
| **BUN residual* (per 1 mg/dL)** | -0.02 (-0.03, -0.01) | <0.001 | -0.03 (-0.04, -0.02) | <0.001 | -0.03 (-0.04, -0.02) | <0.001 | -0.03 (-0.04, -0.02) | <0.001 |

*BUN residual was calculated as the vertical distance from each observation point to the regression fitted curve between baseline BUN and eGFR levels

Model 1: Unadjusted

Model 2: Adjusted for age, sex, smoking, SBP, DM, HTN, CCI

Model 3: Model 2 + adjustment for eGFR, log UPCR, WBC, platelet, log CRP, albumin, total cholesterol, log ferritin, log hepcidin, iron, and TSAT

Model 4: Model 3 + adjustment for use of iron replacement therapy, ESA, ACEi or ARB, statin, and diuretics.

BUN, blood urea nitrogen; SBP, systolic blood pressure; DM, diabetes mellitus; HTN, hypertension; CCI, Charlson comorbidity index; eGFR, estimated glomerular filtration rate as determined by the CKD-EPI creatinine equation; WBC, white blood cell; UPCR, urinary protein-to-creatinine ratio; CRP, C-reactive protein; TSAT, transferrin saturation; ESA, erythropoiesis-stimulating agent; ACEi, angiotensin converting enzyme inhibitor; ARB, angiotensin II receptor blocker

furosemide ≤ 20 mg, torasemide ≤ 10 mg; group 5: furosemide > 20 mg), the results were similar in relation to hemoglobin and BUN (fully adjusted model; β -0.03; 95% CI -0.04, -0.03; P <0.001) and BUN residual (β -0.03; 95% CI -0.04, -0.02; P <0.001).

## Characteristics of patients who developed anemia

Out of the 1,169 patients without anemia at baseline, 414 (35.4%) newly developed anemia during a mean follow-up period of 37.5 ± 22.1 months. The baseline clinical characteristics according to anemia development in these patients are shown in Table 3. The patients in the *de novo* anemia group were older (P <0.001) and had more comorbidities such as DM (P <0.001), HTN (P = 0.008), or higher CCI (P <0.001). They also had higher proteinuria levels (P <0.001) and lower albumin levels (P <0.001), but their CRP levels were not different. BUN was higher and eGFR was lower in the *de novo* anemia group. However, both BUN residual (P = 0.017) and DPI (P = 0.039) were lower in the *de novo* anemia group. Extremely lower BUN residual in CKD stage 4 and 5 patients with *de novo* anemia led to a lower mean value of BUN residual and also lower DPI in CKD stage 4 and 5 patients with *de novo* anemia, which led to a lower DPI in this group as a whole (S3 Table). Nonetheless, DPI was strongly associated with BUN residuals in these patients without anemia at baseline (log-transformed 1000 × DPI; β 12.97; 95% CI 9.03, 16.92; P <0.001). In addition, ESA and iron supplements were administered more often to patients of the *de novo* anemia group (P <0.001 for both). Subjects in the *de novo* anemia group received more diuretics (P <0.001). Considering the types and dose of diuretics, the actual dose of the diuretic was not so high (S4 Table).

## BUN and anemia development

The Kaplan-Meier curves (Fig 3A) show that higher BUN quartile groups had a significantly higher cumulative hazard of anemia development (P <0.0001). The multivariable Cox regression analysis showed that as compared to the lowest BUN quartile, the third and the highest BUN quartiles were associated with increased risk of anemia development (3rd *vs.* 1st quartile; HR 1.62; 95% CI 1.09, 2.40; P = 0.017: 4th *vs.* 1st quartile; HR 2.13; 95% CI 1.28, 3.52; P = 0.003; Table 4). When BUN was analyzed as a continuous variable, the HR for every 1 mg/dL increase in the BUN level was 1.02 (95% CI 1.01, 1.04; P = 0.002). Furthermore, the restricted cubic spline curve adjusted by all the confounding variables showed an almost linear positive relation of BUN to the risk of anemia development (Fig 4A). To remove the effect of possible misassignment of subjects using ESA or iron to group without incident anemia, analysis excluding

**Table 3. Clinical characteristics of the study subjects with respect to anemia development.**

| Characteristics | Total (N = 1,169) | Groups | | P-value |
|---|---|---|---|---|
| | | Incident anemia (-) (n = 755) | Incident anemia (+) (n = 414) | |
| Age (mean ± SD) | 51.9 ± 12.3 | 50.8 ± 12.3 | 53.8 ± 12.1 | <0.001 |
| Sex, male, n (%) | 770 (65.9) | 519 (68.7) | 251 (60.6) | 0.005 |
| SBP (mmHg) | 126.5 ± 14.5 | 125.4 ± 14.2 | 128.4 ± 14.9 | 0.001 |
| BMI (kg/m$^2$) | 24.8 ± 3.4 | 24.9 ± 3.4 | 24.7 ± 3.4 | 0.304 |
| DM, n (%) | 251 (21.5) | 137 (18.2) | 114 (27.7) | <0.001 |
| HTN, n (%) | 1115 (95.4) | 711 (94.2) | 404 (97.6) | 0.008 |
| Age adjusted CCI | 2.7 ± 2.1 | 2.3 ± 2.0 | 3.4 ± 2.1 | <0.001 |
| Smoking status, n (%) | | | | 0.242 |
| Never | 597 (51.1) | 376 (49.8) | 221 (53.4) | |
| Current or former | 572 (48.9) | 379 (50.2) | 193 (46.6) | |
| CKD stage, n (%) | | | | <0.001 |
| Stage 1 | 295 (25.2) | 252 (33.4) | 43 (10.4) | |
| Stage 2 | 314 (26.9) | 239 (31.7) | 75 (18.1) | |
| Stage 3a | 224 (19.2) | 141 (18.7) | 83 (20.0) | |
| Stage 3b | 227 (19.4) | 101 (13.4) | 126 (30.4) | |
| Stage 4 | 103 (8.8) | 22 (2.9) | 81 (19.6) | |
| Stage 5 | 6 (0.5) | 0 (0.0) | 6 (1.4) | |
| Cause of CKD | | | | <0.001 |
| GN | 500 (42.8) | 330 (43.7) | 170 (41.1) | |
| DN | 123 (10.5) | 57 (7.5) | 66 (15.9) | |
| Hypertensive nephropathy | 233 (19.9) | 155 (20.5) | 78 (18.8) | |
| ADPKD | 250 (21.4) | 167 (22.1) | 83 (20.0) | |
| Others | 63 (5.4) | 46 (6.1) | 17 (4.1) | |
| Creatinine (mg/dL) | 1.4 ± 0.6 | 1.2 ± 0.5 | 1.7 ± 0.7 | <0.001 |
| eGFR(mL/min/1.73m$^2$) | 66.4 ± 29.1 | 74.8 ± 27.2 | 51.1 ± 26.1 | <0.001 |
| BUN (mg/dL) | 21.2 ± 9.3 | 18.8 ± 7.1 | 25.6 ± 11.1 | <0.001 |
| BUN residual (mg/dL) | -1.0 ± 5.8 | -0.7 ± 4.3 | -1.3 ± 7.8 | 0.017 |
| *DPI, median, (Q1, Q3) (g/kg/day) | 1.01 (0.87, 1.21) | 1.03 (0.87, 1.22) | 0.99 (0.86, 1.15) | 0.039 |
| WBC (×10$^3$/mm$^3$) | 6694 ± 1921 | 6599 ± 1833 | 6866 ± 2062 | 0.028 |
| Platelet (×10$^3$/mm$^3$) | 230 ± 57 | 230 ± 53 | 230 ± 63 | 0.968 |
| Hemoglobin (g/dL) | 14.2 ± 1.3 | 14.6 ± 1.3 | 13.5 ± 1.0 | <0.001 |
| Iron (ug/dL) | 102.3 ± 35.8 | 105.3 ± 35.7 | 96.8 ± 35.3 | <0.001 |
| Ferritin (pmol/L) | 99.3 (55.0, 174.4) | 105.7 (56.2, 175.4) | 89.8 (52.6, 170.7) | 0.112 |
| TSAT (%) | 33.5 ± 12.0 | 34.0 ± 11.9 | 32.6 ± 12.3 | 0.051 |
| Iron deficiency, n (%) | 617 (53.6) | 373 (50.3) | 244 (59.7) | 0.002 |
| ESA use, n (%) | 17 (1.5) | 1 (0.1) | 16 (3.9) | <0.001 |
| Iron supplement, n (%) | 50 (4.3) | 8 (1.1) | 42 (10.2) | <0.001 |
| Albumin (g/dL) | 4.3 ± 0.3 | 4.3 ± 0.3 | 4.2 ± 0.4 | <0.001 |
| Phosphorus (mg/dL) | 3.5 ± 0.5 | 3.4 ± 0.5 | 3.6 ± 0.5 | <0.001 |
| Uric acid (mg/dL) | 6.7 ± 1.8 | 6.5 ± 1.7 | 7.1 ± 1.9 | <0.001 |
| Total CO$_2$ (mmol/L) | 26.8 ± 3.2 | 27.2 ± 3.0 | 26.1 ± 3.3 | <0.001 |
| Sodium (mmol/L) | 140.9 ± 2.2 | 140.8 ± 2.2 | 141.1 ± 2.3 | 0.049 |
| Potassium (mmol/L) | 4.4 ± 0.5 | 4.4 ± 0.4 | 4.6 ± 0.5 | <0.001 |
| Total cholesterol (mg/dL) | 177.7 ± 37.6 | 178.8 ± 36.8 | 175.6 ± 39.0 | 0.168 |
| LDL cholesterol (mg/dL) | 99.9 ± 31.3 | 101.5 ± 31.4 | 96.8 ± 31.0 | 0.015 |
| Triglyceride (mg/dL) | 160.7 ± 102.9 | 156.9 ± 97.6 | 167.4 ± 111.6 | 0.113 |

*(Continued)*

**Table 3.** (Continued）

| Characteristics | Total (N = 1,169) | Groups | | P-value |
|---|---|---|---|---|
| | | Incident anemia (-) (n = 755) | Incident anemia (+) (n = 414) | |
| HDL cholesterol (mg/dL) | 50.9 ± 15.2 | 51.4 ± 15.0 | 50.5 ± 15.4 | 0.147 |
| CRP, median, (Q1, Q3) (mg/L) | 0.6 (0.2, 1.6) | 0.6 (0.2, 1.4) | 0.6 (0.2, 1.7) | 0.593 |
| UPCR (Q1, Q3) (g/g) | 0.3 (0.1, 0.9) | 0.2 (0.1, 0.6) | 0.5 (0.1, 1.5) | <0.001 |
| Hepcidin (ng/mL) | 11.2 (6.1, 19.1) | 11.1 (6.2, 18.4) | 11.6 (5.7, 20.7) | 0.472 |
| **24hr-urine volume, median, (Q1, Q3) (ml/day) | 1850 (1500, 2300) | 1830 (1490, 2265) | 1900 (1550, 2400) | 0.025 |
| ACEi or ARB, n (%) | 1004 (85.9) | 655 (86.8) | 349 (84.3) | 0.249 |
| †Diuretics, n (%) | 238 (20.4) | 132 (17.5) | 106 (25.6) | <0.001 |
| Statin, n (%) | 581(49.7) | 363 (48.1) | 218 (52.7) | 0.134 |

SD, standard deviation; SBP, systolic blood pressure; BMI, body mass index; DM, diabetes mellitus; HTN, hypertension; CCI, Charlson comorbidity index; GN, glomerulonephritis; DN, diabetic nephropathy; ADPKD, autosomal dominant polycystic kidney disease; eGFR, estimated glomerular filtration rate as determined by the CKD-EPI creatinine equation; BUN, blood urea nitrogen; DPI, dietary protein intake; WBC, white blood cell; TSAT, transferrin saturation; ESA, erythropoiesis-stimulating agent; LDL, low-density lipoprotein; HDL, high-density lipoprotein; CRP, C-reactive protein; UPCR, urinary protein-to-creatinine ratio; ACEi, angiotensin converting enzyme inhibitor, ARB, angiotensin II receptor blocker

**Only 872 and **924 patients in whom 24hr-urine were properly collected and DPI could be calculated were included in the analysis.

†Loop or distal tubule diuretics

iron replacement therapy and ESA use in model 4 (model 5) was performed, which showed the association between anemia development and BUN (HR, 1.03, 95% CI 1.01, 1.04; $P <0.001$) similar to model 4 (Table 4). In another way, sensitivity analysis was performed excluding 9 patients in the non-incident anemia group who were prescribed iron replacement

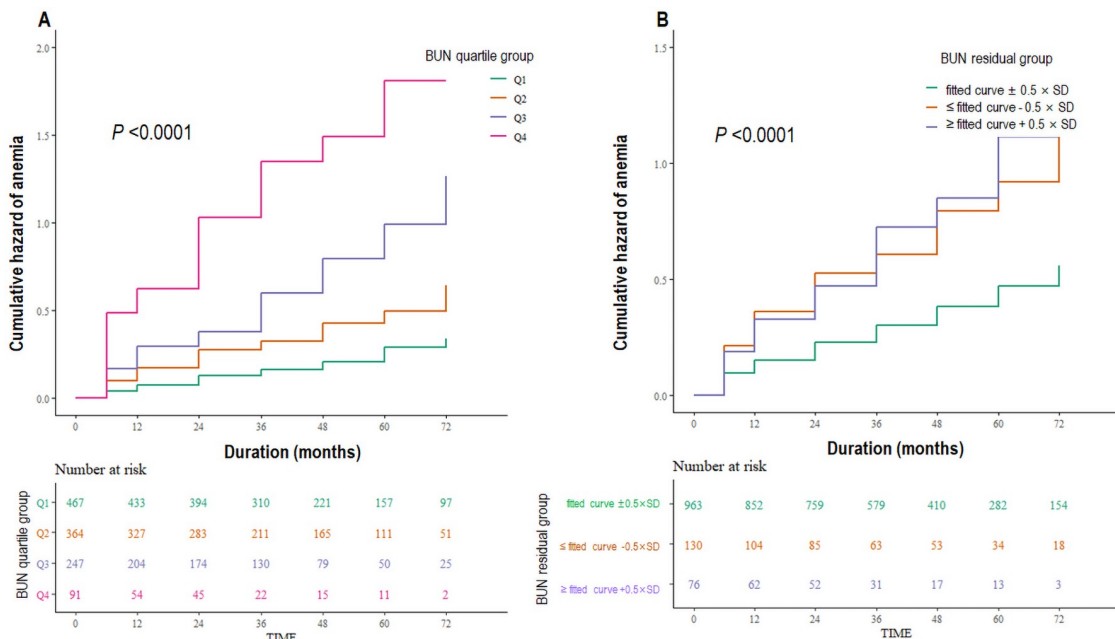

**Fig 3. Kaplan-Meier curves for anemia development according to the BUN quartiles and BUN residual groups. 3A**. The higher BUN quartile groups had a significantly higher cumulative hazard of anemia development ($P < 0.0001$). **3B.** Compared to the group with BUN residual in the fitted curve ± 0.5 × SD, the BUN residual group ≥ fitted curve + 0.5 × SD and ≤ fitted curve– 0.5 × SD had a significantly higher cumulative hazard of anemia development ($P < 0.0001$). BUN, blood urea nitrogen; SD, standard deviation.

**Table 4. Cox regression analysis of the relationship between BUN, BUN residual, and anemia development.**

| Variables | | Model 1 | | Model 2 | | Model 3 | | Model 4 | | Model 5 | |
|---|---|---|---|---|---|---|---|---|---|---|---|
| | | HR (95% CI) | P value | HR (95% CI) | P value | HR (95% CI) | P value | HR (95% CI) | P value | HR (95% CI) | P value |
| **BUN** | Quartile 1 | 1.00 (reference) | | 1.00 (reference) | | 1.00 (reference) | | 1.00 (reference) | | 1.00 (reference) | |
| | Quartile 2 | 1.92 (1.46, 2.52) | <0.001 | 1.22 (0.90, 1.66) | 0.203 | 1.21 (0.87, 1.69) | 0.256 | 1.25 (0.89, 1.76) | 0.193 | 1.22 (0.87, 1.71) | 0.252 |
| | Quartile 3 | 3.37 (2.58, 4.41) | <0.001 | 1.52 (1.06, 2.17) | 0.022 | 1.64 (1.12, 2.41) | 0.011 | 1.62 (1.09, 2.40) | 0.017 | 1.62 (1.10, 2.40) | 0.015 |
| | Quartile 4 | 6.62 (4.81, 9.10) | <0.001 | 2.08 (1.33, 3.24) | 0.001 | 2.33 (1.43, 3.79) | 0.001 | 2.13 (1.28, 3.52) | 0.003 | 2.19 (1.33, 3.60) | 0.002 |
| | Continuous variable (per 1 mg/dL) | 1.05 (1.05, 1.06) | <0.001 | 1.02 (1.01, 1.03) | <0.001 | 1.03 (1.02, 1.04) | <0.001 | 1.02 (1.01, 1.04) | 0.002 | 1.03 (1.01, 1.04) | <0.001 |
| **BUN residual***  | Fitted curve ± 0.5 × SD | 1.00 (reference) | | 1.00 (reference) | | 1.00 (reference) | | 1.00 (reference) | | 1.00 (reference) | |
| | ≤Fitted curve—0.5 × SD | 2.05 (1.59, 2.64) | <0.001 | 1.46 (1.12, 1.90) | 0.005 | 0.90 (0.68, 1.21) | 0.497 | 0.85 (0.63, 1.14) | 0.273 | 0.87 (0.65, 1.17) | 0.345 |
| | ≥Fitted curve + 0.5 × SD | 2.11 (1.52, 2.94) | <0.001 | 1.95 (1.40, 2.72) | <0.001 | 1.82 (1.26, 2.62) | 0.001 | 1.63 (1.11, 2.40) | 0.013 | 1.71 (1.16, 2.50) | 0.006 |
| | Continuous variable (per 1 mg/dL) | 0.99 (0.97, 1.01) | 0.262 | 1.00 (0.99, 1.02) | 0.808 | 1.02 (1.00, 1.04) | 0.017 | 1.02 (1.00, 1.04) | 0.031 | 1.02 (1.00, 1.04) | 0.035 |

*BUN residual was calculated as the vertical distance from each observation point to the regression fitted curve between baseline BUN and eGFR levels

Model 1: Unadjusted

Model 2: Adjusted for age, sex, smoking, SBP, DM, HTN, and CCI

Model 3: Model 2 + adjustment for eGFR, log UPCR, WBC, platelet, log CRP, albumin, total cholesterol, log ferritin, log hepcidin, iron, and TSAT

Model 4: Model 3 + adjustment for use of iron replacement therapy, ESA, ACEi or ARB, statin, and diuretics.

Model 5: Model 3 adjustment for use of ACEi or ARB, statin, and diuretics.

BUN, blood urea nitrogen; HR, hazard ratio; CI, confidence interval; SD, standard deviation; SBP, systolic blood pressure; DM, diabetes mellitus; HTN, hypertension; CCI, Charlson comorbidity index; eGFR, estimated glomerular filtration rate as determined by the CKD-EPI creatinine equation; WBC, white blood cell; UPCR, urinary protein-to-creatinine ratio; CRP, C-reactive protein; TSAT, transferrin saturation; ESA, erythropoiesis-stimulating agent; ACEi, angiotensin converting enzyme inhibitor, ARB, angiotensin II receptor blocker

therapy or ESA at enrollment without adjustment by iron replacement therapy and ESA use. The association between anemia development and BUN (HR, 1.03, 95% CI 1.01, 1.04; $P$ <0.001: model 5) presented similar results. When adjustment was done for the types and dose of diuretics and not just the use of diuretics (group 1: none; group 2: hydrochlorothiazide ≤ 12.5 mg, indapamide ≤ 1.5 mg, metolazone ≤ 2.5 mg; group 3: hydrochlorothiazide > 12.5 mg, indapamide > 1.5 mg, metolazone > 2.5 mg; group 4: furosemide ≤ 20 mg, torasemide ≤ 10 mg; group 5: furosemide > 20 mg), the effect of BUN on anemia development was similar (HR, 1.02: 95% CI 1.01, 1.04; $P$ = 0.002).

## BUN residual and anemia development

The patients were categorized based on 0.5 × SD of BUN from the fitted curve as shown in Fig 2. Kaplan-Meier curves (Fig 3B) show that compared to the group with BUN residual = fitted curve ± 0.5 × SD, the BUN residual group ≤fitted curve—0.5 × SD and ≥fitted curve + 0.5 × SD had significantly higher cumulative hazard of anemia development ($P$ <0.0001). Multivariable Cox regression analysis showed that, compared to the group with BUN residual = fitted curve ± 0.5 × SD, BUN residual group ≥fitted curve + 0.5 × SD was associated with the increased risk of anemia development (HR, 1.63; 95% CI 1.11, 2.40; $P$ = 0.013; Table 4). When

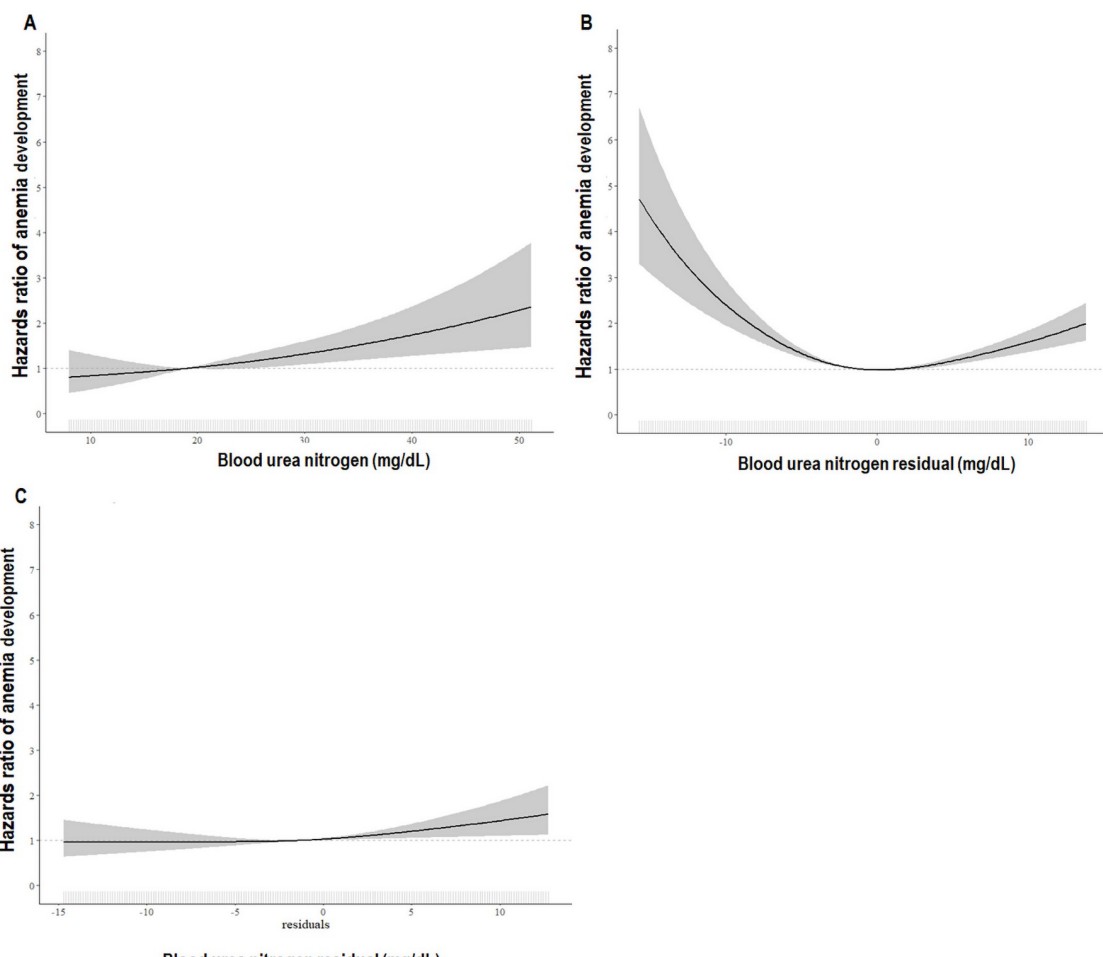

**Fig 4. HRs of anemia development according to BUN and BUN residual. 4A.** Multivariable-adjusted HRs of anemia development according to BUN. A restricted cubic spline curve adjusted by all confounding variables shows almost a linear positive relationship between BUN and the risk of anemia development. The adjusted HR of anemia development was significantly higher at BUN levels >19 mg/dL. Adjusted for BUN, age, sex, smoking, SBP, DM, HTN, CCI, eGFR, log UPCR, WBC, platelet, log CRP, albumin, total cholesterol, log ferritin, log hepcidin, iron, and TSAT, use of iron replacement therapy, ESA, ACEi or ARB, statin, and diuretics. **4B.** Unadjusted HRs of anemia development according to the BUN residual. Consistent with the Kaplan-Meier curve, the unadjusted cubic spline curve shows a U-shaped relationship between the BUN residual and HR of anemia development. **4C.** Multivariable-adjusted HRs of anemia development according to the BUN residual. A restricted cubic spline curve adjusted by all confounding variables shows an almost linear positive relationship between the BUN residual and the risk of anemia development. The adjusted HR of anemia development was significantly higher at BUN residual levels >-1.3 mg/dL. Adjusted for BUN residual, age, sex, smoking, SBP, DM, HTN, CCI, eGFR, log UPCR, WBC, platelet, log CRP, albumin, total cholesterol, log ferritin, log hepcidin, iron, and TSAT, use of iron replacement therapy, ESA, ACEi or ARB, statin, and diuretics. The solid line represents the HRs of anemia, and the shaded areas indicate 95% CI. HR, hazard ratio; SBP, systolic blood pressure; DM, diabetes mellitus; HTN, hypertension; CCI, Charlson comorbidity index; eGFR, estimated glomerular filtration rate as determined by the CKD-EPI creatinine equation; BUN, blood urea nitrogen; WBC, white blood cell; UPCR, urinary protein-to-creatinine ratio; CRP, C-reactive protein; TSAT, transferrin saturation; ESA, erythropoiesis-stimulating agents; ACEi, angiotensin converting enzyme inhibitor, ARB, angiotensin II receptor blocker; CI, confidence interval.

the BUN residual was analyzed as a continuous variable, the HR per 1 mg/dL increase in BUN residual level was 1.02 (95% CI 1.00, 1.04; *P* = 0.031; Table 4). To remove the effect of possible misassignment of subjects using ESA or iron to group without incident anemia, analysis excluding iron replacement therapy and ESA use in model 4 (model 5) was performed, which showed the association between anemia development and BUN residual (HR, 1.02, 95% CI 1.00, 1.04; *P* = 0.035) similar to model 4 (Table 4). In another way, sensitivity analysis was

**Table 5. Cox regression analysis of the relationship between BUN, BUN residual, eGFR, and anemia development according to the stages of CKD.**

| CKD stage | Hemoglobin level (g/dL) | Anemia development, n (%) | Model A | | | | Model B | | | |
|---|---|---|---|---|---|---|---|---|---|---|
| | | | BUN | | eGFR | | BUN residual | | eGFR | |
| | | | HR (95% CI) | P value | HR (95% CI) | P value | HR (95% CI) | P value | HR (95% CI) | P value |
| Stage 1–2 (n = 609) | 14.4 ± 1.3 | 118 (19.4) | 1.04 (0.99, 1.10) | 0.154 | 0.98 (0.97, 1.00) | 0.009 | 1.04 (0.99, 1.10) | 0.154 | 0.98 (0.96, 0.99) | 0.001 |
| Stage 3 (n = 451) | 14.2 ± 1.3 | 209 (46.3) | 1.03 (1.01, 1.05) | 0.010 | 0.99 (0.97, 1.01) | 0.265 | 1.03 (1.01, 1.06) | 0.010 | 0.97 (0.96, 0.99) | 0.007 |
| Stage 4–5 (n = 109) | 13.4 ± 1.1 | 87 (79.8) | 1.00 (0.97, 1.04) | 0.870 | 1.00 (0.93, 1.07) | 0.948 | 1.00 (0.96, 1.03) | 0.872 | 1.00 (0.94, 1.06) | 0.887 |

Model A: Adjusted for BUN, age, sex, smoking, SBP, DM, HTN, CCI, eGFR, log UPCR, WBC, platelet, log CRP, albumin, total cholesterol, Log ferritin, log hepcidin, iron, and TSAT, use of iron replacement therapy, ESA, ACEi or ARB, statin, and diuretics.

Model B: Adjusted for BUN residual instead of BUN in model A.

BUN, blood urea nitrogen; HR, hazard ratio; CI, confidence interval; SBP, systolic blood pressure; DM, diabetes mellitus; HTN, hypertension; CCI, Charlson comorbidity index; eGFR, estimated glomerular filtration rate as determined by the CKD-EPI creatinine equation; WBC, white blood cell; UPCR, urinary protein-to-creatinine ratio; CRP, C-reactive protein; TSAT, transferrin saturation; ESA, erythropoiesis-stimulating agent; ACEi, angiotensin converting enzyme inhibitor, ARB, angiotensin II receptor blocker

performed excluding 9 patients in the non-incident anemia group who were prescribed iron replacement therapy or ESA at enrollment without adjustment by iron replacement therapy and ESA use. The association between anemia development and BUN residual (HR, 1.02, 95% CI 1.00, 1.04; $P = 0.027$: model 5) presented similar results. Similar results were obtained after adjustment for the types and dose of diuretics and not just the use of diuretics (HR, 1.02; 95% CI 1.00, 1.04; $P = 0.036$). Consistent with the Kaplan-Meier curve, the unadjusted cubic spline curve showed a U-shaped relationship between the BUN residual and HR of anemia development (Fig 4B). But the restricted cubic spline curve became rapidly linear as soon as it was adjusted by eGFR, demonstrating an almost linear positive relation of BUN residual with the risk of developing anemia (Fig 4C). In the receiver operating characteristics curves for anemia development, the area under the curve (AUC) value of the BUN residual model (AUC = 0.749) was similar to that of the BUN model (AUC = 0.742).

## Discordant effects of BUN, BUN residual, and eGFR according to the CKD stage

We performed subgroup analysis to show discordant effects of BUN, BUN residual, and eGFR according to the CKD stage (Table 5). BUN, BUN residual, and eGFR were analyzed as continuous variables. In CKD stages 1 and 2, eGFR was associated with the risk of anemia development, while BUN or BUN residual was not. In CKD stage 3, BUN was associated with the risk of anemia development, while eGFR was not. However, eGFR was associated with the risk of anemia development when the BUN residual instead of BUN was used in the same analysis. eGFR, BUN, and BUN residual were not associated with the risk of anemia development in CKD stages 4 and 5.

## Discussion

This study showed that higher BUN was significantly associated with lower hemoglobin levels and an increased risk of *de novo* anemia development in NDCKD patients in models adjusted by multiple confounding variables including eGFR. This association persisted when the BUN residual was used instead of BUN in the same analysis. Furthermore, the discordant effect of

BUN and eGFR on the risk of anemia development was demonstrated in patients with CKD stage 3 in which BUN, not eGFR, was associated with the risk of anemia development. All these findings strongly supported the effects of BUN independent of eGFR on anemia in NDCKD.

BUN is traditionally one of the indicators of kidney function and BUN levels are inversely correlated with GFR [12]. Besides GFR, tubular handling of urea and the production rate of urea are also important determinants of BUN. Consistent with a previous study [17], the scatter plot between eGFR and BUN in this study showed a wide range of BUN concentration at the same eGFR level, which suggested that factors other than eGFR also determine BUN. To eliminate the effect of eGFR on BUN, we obtained the BUN residual [22] from the fitted curve between BUN and eGFR in the scatter plot. As expected, the BUN residual was not related to the eGFR or CKD stage, whereas BUN was related to both eGFR and CKD stage. Importantly, there was no difference in the variables determining BUN and BUN residual except eGFR and CKD stage. Out of the common variables related to BUN and BUN residual, DPI was outstanding in the magnitude of $\beta$ value and statistical significance. Protein intake has been shown to be the main determinant of BUN in previous studies. Urea production showed a linear relationship with serum amino acid concentration [33]. Urea concentration in serum increased when the dietary nitrogen intake increased [34] and BUN levels were widely different depending on the protein intake with the same urea clearance [16]. Lower protein intake reduced BUN in clinical trials involving very low-protein diets in CKD [35]. However, the BUN residual in this study represents DPI relative to eGFR rather than an absolute amount of DPI. Therefore, the same DPI could bring about a higher BUN residual in an individual with lower eGFR and vice versa. A lower eGFR in the higher BUN quartile group might lead to a higher BUN residual despite a similar amount of protein intake between the BUN groups in this study. The larger absolute value of BUN residual in more advanced CKD stage in this study suggested the larger effect of DPI on BUN and BUN residual in subjects with smaller kidney function, which was well demonstrated in a previous study [16]. Therefore, the absolute values of the BUN residual were highly dependent on eGFR although the real numbers of BUN residual were definitely independent on eGFR. The relation of serum phosphorous, uric acid and total $CO_2$ content to BUN and BUN residual might reflect the presence of phosphorous, uric acid, and hydrogen as constituents of protein. Because the patients in the present study were generally stable NDCKD, acute adverse events, such as protein catabolism, dehydration, gastrointestinal bleeding, or severe heart failure scarcely affected BUN or BUN residual. However, it should not be ignored that urine volume and diuretic use also contributed to the variations in BUN and BUN residual by altering tubular handling of urea in this study.

Both BUN and BUN residual were related to baseline hemoglobin and future development of new anemia in this study. Patients with higher BUN might have more production of urea and concomitant nitrogenous uremic toxin than can be excreted by their renal function, which was expressed as higher BUN residual. In addition to EPO deficiency from low eGFR, patients with high BUN might have higher protein-derived uremic toxin(s) hindering effective erythropoiesis, which could explain the effects of high BUN independent of eGFR on anemia in NDCKD in the present study.

Based on the finding that a U-shaped relationship between BUN residual and risk of anemia development in the unadjusted spline curve became linearized when adjusted by eGFR, we performed subgroup analyses according to the CKD stage which showed that not eGFR but BUN increased the risk of anemia development in CKD stage 3. Interestingly, even the unadjusted cubic spline curve between BUN residual and risk of anemia development was almost linear in patients with CKD stage 3. The representation of eGFR, as well as uremic toxin(s) inhibiting effective erythropoiesis by BUN, might make eGFR insignificant in relation to

anemia development in CKD stage 3. Because the BUN residual did not relate to eGFR, eGFR was a significant risk factor for anemia development when the BUN residual was used instead of BUN in the same analysis in CKD stage 3. Because the number of CKD stage 4–5 group patients was smaller than that of other CKD stages, it should be noted that too many variables compared to the number of patients could overly adjust the analysis in these stage groups compared to the number of patients. However, when multivariable correction factors were reduced for analysis (adjusted for BUN (or BUN residual), age, sex, eGFR, UPCR, DM, CCI, log CRP, albumin, TSAT, and ferritin), there was no significant association of BUN, BUN residual, and eGFR with anemia development.

This study raised the importance of adequate protein intake in patients with CKD. While fixed amounts of protein intake were used in all previous studies for the effectiveness of low- or very low-protein diet in retarding renal progression in NDCKD [36, 37], this study suggested that individualized protein intake based on the kidney function rather than an absolute amount of protein intake was important in anemia in NDCKD. A similar concept was actually present with regard to an adequate dose of dialysis. The National Cooperative Dialysis Study demonstrated that patients receiving inadequate dialysis dose to handle their protein intake had a higher failure rate of the attaining the target midweek pre-dialysis BUN [38].

This study is the first study to demonstrate the effects of BUN independent of eGFR on anemia in NDCKD using data from large numbers of NDCKD spanning all CKD stages in a prospective CKD cohort. However, several limitations of this study should be discussed. First, due to observational, not interventional nature of this study, we could not confirm causal relationship between BUN and anemia in our study population. Furthermore, this study could not provide an answer to the question of whether urea is a simple surrogate marker of uremic toxin(s) hindering effective erythropoiesis or direct toxin responsible for the anemia. The effect of urea on erythropoiesis in CKD has been controversial. Urea itself and its metabolic compounds such as cyanate, carbamylated compounds, and ammonia have all been associated with biological changes [39] and linked with atherosclerosis, renal fibrosis, systemic inflammation, and anemia [40–44]. Carbamylated hemoglobin levels were increased by exposure to urea in concentration and time-dependent manners [45]. Another experimental study has shown that carbamylated erythropoietin by cyanate *in vitro* is less biologically active than normal erythropoietin [46]. On the contrary, other study results suggested that urea was not a direct toxin for renal anemia but a surrogate marker for the accumulation of putative uremic toxin(s). The removal of furancarboxylic acid, an inhibitor of erythropoiesis, by albumin-leaking HD improved anemia irrespective of BUN level [9]. A high flux HD improved renal anemia compared with low flux HD with similar Kt/V urea [47]. It was the reduction of parathyroid hormone consequent to better phosphorous control which reduced EPO requirement in CKD patients randomized to a very low-protein diet [19]. In a previous study, higher phosphorous levels (>3.5 mg/dL) were associated with a greater risk for anemia in early CKD stages which suggests that phosphorous may play a role in hemopoiesis [48]. In this regard, it was worthy of noting that phosphorus was also related to BUN in our study and FGF23, the main phosphaturic hormone, was reported to be related to anemia in NDCKD [49, 50]. Second, because this study was performed on a Korean cohort, extrapolations of our findings to other ethnicities should be approached cautiously. The proportion of red meat [51], or plant protein [52], which have been reported to play some role in renal progression would be different among diverse ethnic groups. Different usage of medication among different countries should be considered. The inhibitors of renin-angiotensin system which was associated with significant decrease in hemoglobin concentration [53] was prescribed in 85.3% of this study subjects even though BUN was not

related to this drug category in analysis. Lastly, because BUN residuals were derived from the distributions of eGFR and BUN of our study subjects, the value of BUN residuals would be different depending on the characteristics of subjects of different studies. However, the serial changes in BUN/eGFR ratio in stable NDCKD patients without intercurrent events that could alter tubular handling of urea and/or protein catabolism could be used to guess the changes in protein intake of specific patients.

In conclusion, higher BUN levels were significantly associated with low hemoglobin levels and increased risk of development of *de novo* anemia independent of eGFR in CKD patients. Our results suggested the importance of an adequate amount of protein intake relative to eGFR in individual NDCKD patients.

## Supporting information

**S1 Table. Types and dose of diuretics according to BUN levels.**
(DOCX)

**S2 Table. Multivariable linear regression analysis for related factors to BUN and BUN residual.**
(DOCX)

**S3 Table. BUN residual and DPI according to CKD stages.**
(DOCX)

**S4 Table. Types and dose of diuretics with respect to anemia development.**
(DOCX)

## Author Contributions

**Conceptualization:** Hyo Jin Kim, Curie Ahn, Kook-Hwan Oh, Dong-Wan Chae.

**Data curation:** Hyo Jin Kim, Tae Eun Kim, Miyeun Han, Yongin Yi, Jong Cheol Jeong, Ho Jun Chin, Sang Heon Song, Joongyub Lee, Kyu-Beck Lee, Suah Sung, Seung Hyeok Han, Eun Young Seong, Kook-Hwan Oh, Dong-Wan Chae.

**Formal analysis:** Hyo Jin Kim, Tae Eun Kim, Joongyub Lee.

**Funding acquisition:** Curie Ahn.

**Investigation:** Miyeun Han, Yongin Yi, Jong Cheol Jeong, Ho Jun Chin, Sang Heon Song, Kyu-Beck Lee, Suah Sung, Seung Hyeok Han, Eun Young Seong, Curie Ahn, Kook-Hwan Oh.

**Methodology:** Hyo Jin Kim, Tae Eun Kim, Jong Cheol Jeong, Ho Jun Chin, Sang Heon Song, Joongyub Lee, Kyu-Beck Lee, Suah Sung, Seung Hyeok Han, Eun Young Seong, Curie Ahn, Kook-Hwan Oh, Dong-Wan Chae.

**Project administration:** Curie Ahn.

**Supervision:** Eun Young Seong, Curie Ahn, Kook-Hwan Oh, Dong-Wan Chae.

**Validation:** Kook-Hwan Oh.

**Writing – original draft:** Hyo Jin Kim.

**Writing – review & editing:** Hyo Jin Kim, Dong-Wan Chae.

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
