## [Decision Letter · Decision Letter 0]

10 Mar 2021

PONE-D-20-41118

Effects of blood urea nitrogen independent of the estimated glomerular filtration rate on the development of anemia in non-dialysis chronic kidney disease : The results of the KNOW-CKD study

PLOS ONE

Dear Dr. Dong Wan Chae,

Thank you for submitting your manuscript to PLOS ONE. After careful consideration, we feel that it has merit but does not fully meet PLOS ONE’s publication criteria as it currently stands. Therefore, we invite you to submit a revised version of the manuscript that addresses the points raised during the review process.

We look forward to receiving your revised manuscript.

Kind regards,

Ping-Hsun Wu, M.D. PhD.

Academic Editor

PLOS ONE

Additional Editor Comments (if provided):

More information on BUN residual is needed in the introduction section. The criteria of ESA prescription should be mentioned in the method section. The medications or conditions that potentially influence the BUN level should be considered in this study.

Journal Requirements:

2)  Please provide a sample size and power calculation in the Methods, or discuss the reasons for not performing one before study initiation.

3) Please note that PLOS does not permit references to “[data] not shown.” Authors should provide the relevant data within the manuscript, the Supporting Information files, or in a public repository. If the data are not a core part of the research study being presented, we ask that authors remove any references to these data.

4) We noticed you have some minor occurrence of overlapping text with the following previous publication(s), which needs to be addressed:

- https://centerhealthyminds.org/assets/files-publications/goldberg-still-facial-photographs.pdf

- https://bmcnephrol.biomedcentral.com/articles/10.1186/s12882-018-1005-3

- https://www.nature.com/articles/s41598-019-52499-6

In your revision ensure you cite all your sources (including your own works), and quote or rephrase any duplicated text outside the methods section. Further consideration is dependent on these concerns being addressed.

Reviewers' comments:

Reviewer's Responses to Questions

**Comments to the Author**

1. Is the manuscript technically sound, and do the data support the conclusions?

Reviewer #1: Partly

Reviewer #2: Partly

2. Has the statistical analysis been performed appropriately and rigorously? 

Reviewer #1: Yes

Reviewer #2: No

3. Have the authors made all data underlying the findings in their manuscript fully available?

Reviewer #1: Yes

Reviewer #2: Yes

4. Is the manuscript presented in an intelligible fashion and written in standard English?

Reviewer #1: Yes

Reviewer #2: Yes

5. Review Comments to the Author

Reviewer #1: In this article, the authors investigated the effects of BUN on anemia in non-dialysis CKD patients. By obtaining the BUN residual from the fitted curve between BUN and eGFR, they found that high BUN residual level was associated with anemia development. The manuscript is well written and easy to follow and the study has novelty. However, some queries need to be clarified before making the conclusion.

1. What is the clinical meaning of the BUN residual? The authors need to expand more details of BUN residual in the section of INTRODUCTION.

2. The type and dose of diuretics are the major concern in this study. Higher diuretic dose would change the levels of BUN and BUN residual easily. How did the authors adjust this variable?

3. BUN residual is not easy to get during daily medical practice. If physicians cannot get the BUN residual easily, how to expand its clinical usage?

4. Table 5 dose not list the reference group.

Reviewer #2: Kim et al. conducted a prospective cohort study demonstrating that BUN and BUN residuals predict the development of anemia in non-dialysis CKD patients. Although the article is interesting, a few questions were raised and should be further addressed.

1. In the Introduction, there is a sentence “no specific uremic toxin(s) responsible for anemia of 90 CKD has not been identified yet”. I guess “not” is redundant.

2. Although mentioned in the main text of Results, the proportions of patients receiving iron therapy or ESA should be added in Tables 1 and 2.

3. Model 4 were adjusted for several additional variables including ESA. What is the criteria of ESA prescription? According to KDIGO guideline 2012 for anemia, 3.4.1: For adult CKD ND patients with Hb concentration >= 10.0 g/dl, we suggest that ESA therapy NOT be initiated. (2D) A lower Hb should be detected prior to ESA prescription, suggesting that the primary outcome, the development of anemia (Hb <13 for men and Hb <12 for women), occurred prior to ESA prescription. In this condition, why ESA is a concern and required to be adjusted?

4. Another important issue is that the primary outcome, the occurrence of anemia, is a condition confounded by renal function at that time. As shown in Table 3, patients having incident anemia had higher Cre, lower eGFR, and higher CKD stages. To avoid this confounding, the authors should, for example, look at the proportion of anemia or Hb levels when entering CKD stage 4 or 5, or adjust for renal function when the primary outcome occurred (not simply adjusted for the baseline renal function).

5. The sentence below the subtitle of “BUN residual and anemia development”, I think it’s “0.5 x SD of BUN” rather than “0.5 x SD of BUN residual”?

6. Is there statistical evidence to support the use of 0.5 SD as a criterion of fitting the curve? or it’s simply empirical?

7. The problems of discordant effects of BUN, BUN residuals, and eGFR as per the CKD stage shown in Table 5 were again confounded by renal function at the time of anemia developed.

8. The authors used single measurement of BUN for analysis; however, several conditions may influence BUN levels, particularly daily protein intake and a relatively dehydration status after more than 8-hour fasting for the blood test. Did the authors consider to average the BUN levels within a given period of time for a better representation of the actual situation?

6. PLOS authors have the option to publish the peer review history of their article (what does this mean?). If published, this will include your full peer review and any attached files.

Reviewer #1: No

Reviewer #2: No

---

## [Author Response · Author response to Decision Letter 0]

5 Apr 2021

We uploaded the response letter to the file.

---

## [Decision Letter · Decision Letter 1]

3 Jun 2021

PONE-D-20-41118R1

Effects of blood urea nitrogen independent of the estimated glomerular filtration rate on the development of anemia in non-dialysis chronic kidney disease : The results of the KNOW-CKD study

PLOS ONE

Dear Dr. Dong Wan Chae,

Thank you for submitting your manuscript to PLOS ONE. After careful consideration, we feel that it has merit but does not fully meet PLOS ONE’s publication criteria as it currently stands. Therefore, we invite you to submit a revised version of the manuscript that addresses the points raised during the review process.

We look forward to receiving your revised manuscript.

Kind regards,

Ping-Hsun Wu, M.D. PhD.

Academic Editor

PLOS ONE

Journal Requirements:

Additional Editor Comments (if provided):

There are still several excellent comments and suggestions from Reviewer 2. ESA and iron treatment are important issues for analysis in this study. Besides, the statistical methods to fit the curve between eGFR and BUN should be described clearly. Please revise accordingly.

Reviewers' comments:

Reviewer's Responses to Questions

**Comments to the Author**

1. If the authors have adequately addressed your comments raised in a previous round of review and you feel that this manuscript is now acceptable for publication, you may indicate that here to bypass the “Comments to the Author” section, enter your conflict of interest statement in the “Confidential to Editor” section, and submit your "Accept" recommendation.

Reviewer #1: All comments have been addressed

Reviewer #2: (No Response)

2. Is the manuscript technically sound, and do the data support the conclusions?

Reviewer #1: Yes

Reviewer #2: Partly

3. Has the statistical analysis been performed appropriately and rigorously? 

Reviewer #1: Yes

Reviewer #2: Yes

4. Have the authors made all data underlying the findings in their manuscript fully available?

Reviewer #1: Yes

Reviewer #2: Yes

5. Is the manuscript presented in an intelligible fashion and written in standard English?

Reviewer #1: Yes

Reviewer #2: Yes

6. Review Comments to the Author

Reviewer #1: The authors have answered all the questions well, including the e information on BUN residual and the types and dose of diuretics. I have no further comments.

Reviewer #2: Kim et al. conducted a study evaluating the effects of BUN (and its residual) on anemia in non-dialysis CKD patients. Although interesting, a number of questions and concerns remained unanswered.

1. In the Method section, the statistical methods (or equations) that used to fit the curve between eGFR and BUN should be disclosed, since BUN residual is an important factor discussed throughout the article.

2. To better illustrate the dose-dependent effects of BUN on variables in Table 1, did the authors consider the p-value for trend instead of just p-value?

3. Based on the description in the article, ESA was prescribed in patients with renal anemia with hemoglobin < 10 g/dL in accordance with the KDIGO guidelines. The authors did not disclose the indication of iron supplementation. Besides iron deficiency based on iron profile, we usually prescribe iron only when Hb falls below certain level. In my opinion, those patients on ESA and iron treatment should still be removed from the second part of analysis because their Hb levels had already dropped earlier and against the notion to detect de novo anemia. The patients remained non-anemic by definition during follow-up could simply be owing to escalated dose of ESA and iron, which could possibly offset the effect of increased BUN. If the authors decided not to change the current analysis, at least the description in the responding to reviewers’ comments “we performed the analysis with adjustment excluding ESA use in which the association between anemia development and BUN (HR, 1.02, 95% CI 1.01, 1.04; P=0.001) or BUN residual (HR, 1.02, 95% CI 1.00, 1.04; P=0.027) showed similar results to the previous analysis” should be added in the article.

4. As disclosed in the Methods, the study cohort enrolled about 600 subjects with polycystic kidney disease. How many patients had polycystic kidney disease in this study? Anemia is on average less severe in ADPKD patients.

5. In a paragraph evaluating the dose of diuretics, which kind of thiazide did the authors refer to? Thiazides actually represent a group of agents, such as hydrochlorothiazide and chlorothiazide. And according to WHO Collaborating Centre for Drug Statistics Methodology (https://www.whocc.no/atc_ddd_index/?code=C03BA&showdescription=no), the DDDs for indapamide and metolazone are 2.5 mg and 5 mg. What’s the rationale of 1.5 mg for indapamide and 2.5 mg for metolazone in groups 2 and 3?

6. Is it possible that low BUN residual and low DPI in patients with de novo anemia shown in Table 3 simply suggested that these patients had malnutrition?

7. In Table 5, I am not sure how the statistical process was performed. Did the authors repeat the regression analysis in each CKD stage for the effects of BUN and BUN residual on anemia development? If so, there’s only 109 patients in CKD stage 4-5, and over-adjustment for variables is a concern.

8. In the Discussion, a sentence goes “The National Cooperative Dialysis Study demonstrated that patients with higher midweek pre-dialysis BUN resulting from inadequate dialysis dose to handle their protein intake had a higher failure rate [38].” What kind of failure did the authors refer to?

7. PLOS authors have the option to publish the peer review history of their article (what does this mean?). If published, this will include your full peer review and any attached files.

Reviewer #1: No

Reviewer #2: No

---

## [Author Response · Author response to Decision Letter 1]

15 Jul 2021

We uploaded the file to respond to the reviewer.

---

## [Decision Letter · Decision Letter 2]

31 Aug 2021

Effects of blood urea nitrogen independent of the estimated glomerular filtration rate on the development of anemia in non-dialysis chronic kidney disease : The results of the KNOW-CKD study

PONE-D-20-41118R2

Dear Dr. Dong Wan Chae,

We’re pleased to inform you that your manuscript has been judged scientifically suitable for publication and will be formally accepted for publication once it meets all outstanding technical requirements.

Kind regards,

Ping-Hsun Wu, M.D. PhD.

Academic Editor

PLOS ONE

Additional Editor Comments (optional):

All suggestions had been revised accordingly. This manuscript is available for publication.

Reviewers' comments:

Reviewer's Responses to Questions

**Comments to the Author**

1. If the authors have adequately addressed your comments raised in a previous round of review and you feel that this manuscript is now acceptable for publication, you may indicate that here to bypass the “Comments to the Author” section, enter your conflict of interest statement in the “Confidential to Editor” section, and submit your "Accept" recommendation.

Reviewer #1: All comments have been addressed

Reviewer #2: All comments have been addressed

2. Is the manuscript technically sound, and do the data support the conclusions?

Reviewer #1: Yes

Reviewer #2: Yes

3. Has the statistical analysis been performed appropriately and rigorously? 

Reviewer #1: Yes

Reviewer #2: Yes

4. Have the authors made all data underlying the findings in their manuscript fully available?

Reviewer #1: Yes

Reviewer #2: Yes

5. Is the manuscript presented in an intelligible fashion and written in standard English?

Reviewer #1: Yes

Reviewer #2: Yes

6. Review Comments to the Author

Reviewer #1: The authors have comprehensively answered all the questions of reviewer 2 . I have no further comments.

Reviewer #2: The authors have addressed all the comments. There is no more question from my side. Thank you for your time and all the efforts.

7. PLOS authors have the option to publish the peer review history of their article (what does this mean?). If published, this will include your full peer review and any attached files.

Reviewer #1: No

Reviewer #2: No

---

## [Editor Report · Acceptance letter]

3 Sep 2021

PONE-D-20-41118R2 

Effects of blood urea nitrogen independent of the estimated glomerular filtration rate on the development of anemia in non-dialysis chronic kidney disease: The results of the KNOW-CKD study 

Dear Dr. Chae:

I'm pleased to inform you that your manuscript has been deemed suitable for publication in PLOS ONE. Congratulations! Your manuscript is now with our production department. 

Kind regards, 

on behalf of

Dr. Ping-Hsun Wu 

Academic Editor

PLOS ONE